# Use of Slag from the Combustion of Solid Municipal Waste as A Partial Replacement of Cement in Mortar and Concrete

**DOI:** 10.3390/ma13071593

**Published:** 2020-03-31

**Authors:** Monika Czop, Beata Łaźniewska-Piekarczyk

**Affiliations:** 1Department of Technologies and Installations for Waste Management, Faculty of Energy and Environmental Engineering, The Silesian University of Technology, Konarskiego 18, 44-100 Gliwice, Poland; 2Department of Building Engineering and Building Physics, Faculty of Civil Engineering, The Silesian University of Technology, Akademicka 5, 44-100 Gliwice, Poland; beata.lazniewska@polsl.pl

**Keywords:** municipal solid waste minimization, incineration, circular economy, slag, addition, cement, mortar, concrete

## Abstract

In Europe, the use of wastes in the cement and construction industry follows the assumptions of sustainability and the idea of circular economy. At present, it is observed that cement plants introduce wastes to the cement in the form of so-called mineral additives. The most often used mineral additives are: fly ash with silica fume, granulated blast furnace slag and silica fume. The use of mineral additives in the cement is related to the fact that the use of the most expensive component of cement—Portland cement clinker—is limited. The purpose of the article is a preliminary evaluation of the suitability of slag from the municipal solid waste incineration plant for its use as a replacement of cement. In this article, slag from the municipal solid waste incineration (MSWI) replaces cement in the quantity of 30%, and presents the content of oxides and elements of slag from the MSWI. The obtained results are compared to the requirements that the crushed and granulated blast furnace slag need to meet to be suitable for use as an additive of type II to the concrete. The conducted analyses confirmed that the tested slag meets the requirements for the granulated blast furnace slag as an additive to the concrete in the following parameters: CaO ≤ 18.0%, SO3 ≤ 2.5% and Cl ≤ 0.1%. At the same time, mechanical features were tested of the designed mortars which consisted of a mixture of Portland cement (CEM I) with 30% of slag admixture. The designed mortar after 28 days of maturing reached a compressive strength of 32.0 MPa, and bending strength of 4.0 MPa. When compared to the milled granulated blast furnace slag (GBFS), the obtained values are slightly lower. Furthermore, the hardened mortars were subject to a leachability test to check the impact on the environment. Test results showed that the aqueous extracts from mixtures with 30% of slag admixtures slightly exceed the limits and do not pose a sufficiant threat to the environment as to eliminate the MSWI slag from economical use.

## 1. Introduction

In the past couple of years, we have observed a constant increase of the produced municipal waste. In 2018, 2.5 billion Mg of waste was generated in the EU countries. Municipal waste accounts for about 10% of their total stream. According to the data provided by Eurostat [1], in 2019, each inhabitant of Europe produced 489 kg of municipal wastes. Municipal wastes are considered as a global problem because they are very much visible and have very complex characteristics. Their amount and morphological content depend strongly on the civilization development, quality of life and society’s wealth. The form and weight of the produced municipal wastes are also strongly affected by the population density, type of land development (single houses, multi-family buildings), touristic attractiveness, public utility buildings and type, size and amount of commercial buildings [2].

The increasing stream of municipal wastes becomes one of the most important challenges in the European Union in respect of the environment protection and legal requirement. The waste directive 2008/98/EC explicitly specifies the order of importance in waste management: reduce, reuse, recycle, recover and dispose of [3]. The strategic goal of the EU is a complete elimination of the disposal of municipal wastes. The municipal wastes, first of all, need to be recycled (material, chemical and organic recycling) or, as an alternative, combusted with the purpose of the energy recovery. Energetic use of waste is a good solution for poor quality fractions. These wastes, due to their high heterogeneity and amount of contaminants, cannot be directed to material, raw material or chemical recycling. Until now, they “ended their life cycle” at the landfill. This solution is a waste of resources. Energetic use brings great ecological benefits, which are associated with reducing the amount of waste deposited. In addition to ecology, economics is important. Economic benefits are associated with saving exhaustible resources of fossil fuels.

One of the methods of solid waste management is thermal degradation in a municipal solid waste incineration plant (MSWI). It needs to be noted that the wastes that undergo thermal processing are the wastes which, after selective collecting, for many reasons are not suitable for recycling (i.e., low quality, high contamination, non-homogenous stream) but that have high energy value. The thermal degradation of wastes is based on the possibility to use the wastes to obtain heat or electrical energy. Furthermore, it reduced up to 90% the volume of the stream of wastes. As a result of the thermal processing, secondary solid waste is produced [2,4]. Solid secondary wastes include fly ash solid waste from the purification of the exhaust fumes and bottom slag. Particular attention shall be paid to the bottom slag, which following preliminary valorization may constitute valuable alternative aggregate for the use of the construction industry [5].

Correctly conducted process of combustion/co-combustion of wastes should guarantee an appropriate level of transformation of solid products of combustion, expressed by maximum content of non-oxygenated organic compounds in the following measurements [6]:The total content of organic carbon in slags and furnace ash should not exceed 3%.The percentage of combustible elements in slags and furnace ash should be less than 5%.

The use of the secondary products in the cement and construction industry is compliant with the idea of the circular economy (Figure 1) [5]. Replacing clinkers with active mineral additives starts to play a more and more important role in the production technology of the types of cement. Worldwide, in the production of cement, this process is based both on economic and ecological values.

Poland does not have any experience in the use of this sort of waste in the construction industry, because in previous years (up to 2016) there were not any significant amounts of these wastes. We only base on Poland’s experience, within an academic register of foreign plants which cannot be transferred to the Polish market. It needs to be noted that the quality of the input stream to the system reflects the output stream. At present, Poland is at the stage of constructing a modern waste management system with the elements of the circular economy. Therefore, we need to conduct tests and broaden our knowledge on the changeability of the input stream in the course of the time depending on the thermal processing and evaluate the impact of the thermal processing on the composition of the secondary materials (slag, fly ash, waste from the purification of the exhaust fumes). Such information will be helpful, if not indispensable, when choosing the best criteria for the safe use of the secondary materials in the construction industry, avoiding any negative impact on the environment. A potential threat when using secondary waste from the thermal degradation of municipal waste for the production of mortar and concrete is the migration of heavy metals to the soil and water environment. High levels of chloride salts and sulfates may prove to be environmentally problematic. The negative effect of salt migration into the environment may be the extinction of living organisms, which can lead to a reduction in species biodiversity [7]. The high salt content can inhibit plant growth, reduce their size and the number of leaves and roots. As a result of salt accumulation in soil, nutrients such as phosphorus, calcium and potassium may be leached.

The approach presented in this article is ecologically friendly because it aims at the best and safe use of the slag from the MSWI by replacing part of the cement with slag of similar parameters. In this way, the emission of the greenhouse gases in cement plants, mainly emission of CO_2_ can be limited. It needs to be noted that with the production of 1 Mg of cement 1 Mg of CO_2_ is generated and the participation of the cement industry in the global CO_2_ emission caused by human being amounts to 5%–8% [8].

## 2. The Analyzed Municipal Solid Waste Incineration Plant

The thermal degradation installation processes 210 Mg of waste annually. It has two combustion lines that work independently of each other. The boiler has a temperature of 1000 °C, and cleaned exhaust gases are discharged up to a 50 m chimney. The final effect of the process is electricity (128,000 MWh/year) and thermal energy (300,000 GJ/year). The energy produced is used for the needs of the city [9].

The installation accepts waste with an average calorific value of 7.5 GJ/Mg. These include, among others: residua waste (20 03 01), bulky waste (20 03 07), in addition to waste constituting combustible waste (refuse-derived fuel, 19 12 10) [10].

Residual municipal waste collected in the city is transported by cars to the installation. The vehicle enters the scale. The mass of collected waste is measured and passes through the radiosensitive gate. The vehicle is directed to the delivery hall. The hall has negative pressure to prevent unpleasant odours from spreading. The waste goes to the bunker, in which the gripper mixtures the waste to avoid self-ignition and to unify the mass in the event of installation shutdown. An independent deodorizing system is installed in the bunker. Pre-prepared waste is ready for thermal treatment with energy recovery. The waste is directed to the hopper using a gripper. Then the feeder is directed to the grate, where they are thermally transformed. The waste gases generated in the process are cleaned [9]. The flue gas cleaning system in the installation consists of the following elements: bag filters, NOx reduction system, semi-dry reactor, draft fan, chimney and emission measurement station, turbine-generator, air-cooled condenser and heat exchanger, which are installed for energy recovery. Secondary wastes from the process are: slag (19 01 11), fly ash (19 01 13 *) and solid waste from flue gas treatment (19 01 07 *) [9]. A scheme depicting the process of solid waste management is presented in Figure 2.

## 3. MSWI Slag as A Potential Component of Cement for Mortar and Concrete

Analysis of the mechanical properties indicates that slag (S) from the MSWI plant is a component that fills microstructure of mortars without pozzolan properties, unlike granulated blast furnace slag (GBFS). When preparing mortars based on MSWI slag, particularly undesirable are reactions between cement and aluminum and zinc found in the wastes. Concrete may swell because of the salts such as sulfurs (SO_4_), which can be found, for example, in the soil, in the direct contact with the concrete. It leads to a chemical reaction with aluminum contained in the cement generating an expansive substance. These phenomena may lead to inciting internal tensile forces and consequently to the occurrence of cracks, in case of excessive concrete resistance [11,12,13]. The reasons and the process of the above mentioned chemical reactions are described in details in the publications. It is known that these reactions cause swelling and even cracking of concrete [12,13,14].

Furthermore, non-homogeneity of the slag from the MSWI plant and significant content of harmful components (unburnt coal and sulfur compounds) are the main reasons for low resistance of the concrete and limit the way of its use. Depending on the composition, concrete strength reaches 2–10 MPa. Slag from MSWI plant shall not be used for the production of reinforced concretes and concretes exposed to permanent humidity over 75%. When the amount of the substance mentioned above is high, mortar/concrete may tend to change volume (shrinkage, swelling) under the wet conditions. The reason is usually the content of unburnt coal. Water absorption of the slag-concrete, depending on the type of slag and composition of concrete, equals 15%–25%. Frost resistance is generally satisfactory, under the condition that the slag used does not contain a large amount of unburnt coal (C < 5%), and the concrete resistance is not lower than 5 MPa [13,14].

Concrete with the additive of slag from the MSWI is used in the building industry almost exclusively for the production of various types of hollow wall bricks and less often for the construction of monolith walls in the first two floors in individually constructed houses. Due to the unfavorable results of experiments, it is no longer used in the monolith wall, and only small wall elements are made of it.

To be suitable for use in a responsible way, the slag from the MSWI plant needs to be seasoned and processed. Depending on the original composition and eventual destination, processing of slag may assume various forms (seasons, rinsing, washing, exposing to sodium hydroxide, removal of heavy metals, vitrification to assure the expected quality of the concrete) [11]. One of the most frequent problems will be the necessity to reduce aluminum and the amount of glass contained in the slag and control, among other elements such as the amount of unburnt coal, chloride, zinc and sulfur. Additionally, the possibility that potentially hazardous substances get through to the environment needs to be limited. [14,15].

There is a lack of national requirements for the slag of the MSWI as the addition to the cement. For the needs of this article, the discussions were based on the requirements for the milled granulated blast furnace slag (GBFS), used as additive of type II to concrete [16]. The standard requirements for GBFS are presented in Table 1.

The purpose of the article is a preliminary evaluation of the physical, chemical and mechanical properties of the slag from the municipal solid waste incineration plant used as an addition to the mortar and in future in concrete. The volume of the mortar in concrete is about 55%–60%, which allows the assessment of the concrete based on the examination of the properties of the mortar (they do not take into account the influence of the coarse aggregate compact zone).

## 4. Materials and Methods

### 4.1. Materials

For the tests, Portland cement was used-CEM I 42.5R (further on marked as CEM I) following the requirements of PN-EN 197-1:2002 [17]. The main component of the Portland cement CEM I (Figure 3a) is Portland cement clinker (≥ 95%). The material was a product of sintering of the mixture of raw materials. Its main chemical components are four oxides: calcium oxide, silicon oxide, aluminum oxide and iron oxide.

The second material used for tests was slag (Figure 3b) which was generated as a side effect of the thermal processing of municipal wastes. Within a year in the MSWI plant, which was the time-span of consideration, 55 909 Mg of coarse and fine slag was produced [9].

At the beginning of the research, a chemical analysis of the tested materials was carried out. Oxides content and concentration of heavy metals in cement and slag from the MSWI were investigated. The obtained results are presented in Table 2, Table 3 and Table 4. The tested materials have similar oxides content. The basis phase component of slag from the MSWI is SiO_2_. The high content of silica (SiO_2_ >50%) may be reflected in respective high pozzolana activity. Content of CaO meets the requirements for the crushed, granulated blast furnace slags. (CaO ≤ 18%).

The composition of the tested mortars is presented in Table 4. Mortar containing CEM I 42.5 was prepared according to requirements of EN-196-1 [17]. Portland cement CEM I 42.5R (marked as CEM I) was used in case of reference mortar. The binder made with Portland cement CEM I 42.5R with 30% of slag from the incineration plant (hereinafter marked as CEM I + 30% S-MSWI) was used to investigate the influence of slag (MSWI) to properties of mortars. The binder made with the Portland cement CEM I 42.5R with 30% the addition of crushed blast furnace gravel (hereinafter marked as CEM I + 30% GBFS) was used as reference slag to MSWI slag.

The mortars were prepared following the procedure described in standard EN 197-1:2011 [18] with the use of automatic vortex mixer (Table 3).

Conducted tests of heavy metals content in cement CEM I confirmed a high level of most of the trace elements, i.e., Zn > Cr > Cu > Pb > V > Ni > As > Cd > Tl > Hg (Table 4). Higher content of heavy metals in CEM I is related to the increased content of waste fuels in the production of clinker. According to the author of publication [19,20], at present, it is difficult to determine the source of heavy metals in the cement. However, the content of heavy metals in the dry mass of tested slag was high. The raw of elements was as follows: Cu > Zn > Pb > Cr > Ni > V > As > Cd > Tl > Hg.

### 4.2. Methods

The testing procedure is planned and realized in such a way as to determine the characteristics of slag (physical and chemical) which are important in the light of its use a partial replacement of cement. Impact on the environment was also taken into consideration.

Moisture content in the tested sample of slag was determined in compliance with the standard EN 15934: 2013 [21] and bulk density was determined following standard EN 1097-3:2000 [22].

The slag sample was crushed in a ball mill and underwent physical and chemical analyses. The specific surface was determined according to standard EN 196-6:2019-01 [23] and loss at ignition following standards: EN 15935:2013-02 [24] and EN 196-2:2013-11 [25]. Slag sample was also tested for the content of the following elements: carbon (C)-EN 15407:2011 [26], organic carbon (TOC)-EN-Z-15011-3:2001 [27], sulfur (S)-EN-ISO 334:1997 [28] and chloride (Cl)-EN-ISO 587:2000 [29].

The concentration of sodium, calcium, potassium, lithium and barium was determined with the flame photometry method—in compliance with the standard EN-ISO 9964-3:1994 [30].

To determine the concentration of heavy metals in the dry mass of samples, an inductively coupled plasma mass spectrometer from Perkin Elmer Company was used. It allows defining elements activated in argon plasma [31].

The aqueous extract from the slag was prepared in compliance with the standard PN-EN 12457-2:2006 [32]. From the waste weighing 2 kg, a representative laboratory sample was prepared. Tested wastes were sieved through sieves of 2 mm mesh size. From such a prepared sample aqueous extract was made. The relations of liquid to the solid phase was L/S = 10 dm^3^/kg (basic test). The elution liquid was distilled water of pH 7.1 [33] and electrical conductivity of 61.18 µS/cm [34]. Subsequently, the extract was mixed in the vortex mixer for 24 h, before filtering the suspension. Aqueous extracts for mortar samples were prepared in the same way. The mortar specimens were solid and crushed to the grain size of < 10 mm.

Analysis of the aqueous extracts from slag and cement mortar included several determinations. Content of organic carbon was determined using analyzer Vario TOC Cube from the company Elementar [35]. The pH of the solutions [33] and conductivity [34] was identified with the use of Elmetron CPC-501 apparatus. The concentration of chlorides was determined with the Mohr method with the use of silver nitrate (v) as titration reagent and potassium dichromate (VI) as an index (PN-ISO 9297:1994 [36]). The content of sulphates (VI) (SO_4_^2−^) was tested with gravitation method with barium chloride, according to the standard PN-ISO 9280:2002 [37].

The concentration of potassium, calcium, lithium and barium in aqueous extracts of ashes and mortar was determined with the flame photometry method–in compliance with standard EN-ISO 9964-3:1994 [30]. For the evaluation of the content of heavy metals in the aqueous extract an inductively coupled plasma mass spectrometer was used from Perkin Elmer company, which allows determining elements arising in argon plasma [31].

Binding time of the slurry of CEM I 42.5R with the 30% slag was tested in the automatic device of Vicata according to the procedure specified in EN 196-3 [38].

Consistency of the mortars was determined using a flow table method as specified in EN 1015-3 [39]. The mortar was placed on a flow table in the truncated mould in two layers. After that, the flow table was mechanically raised by 10 mm and dropped at a rate of once per second for about 15 s. The flow diameter of the mortar was measured in orthogonal directions to determine the consistency of mixtures.

Further, the air content was measured as per EN 1015-7:1999 [39] by pouring and tamping the mortar in the air entrainment meter and applying an air pressure that forced the application of water into the mortar and relocating the air within the pores.

Determination of the volume changes according to EN 196-3 [40] is conducted with the use of standard slurry (Table 3), which is placed in Le Chatelier’s device with measurement wires. Once the ring is filled up with the cement paste, it is preserved in the temperature of 20 °C with the relative humidity of 98% for 24 h. After 24 h, the distance X between the wires should be measured. After that, The cement paste leaven in le Chatelier ring is then boiled in water for 3 h. In the next step of research, the ring is cooled to the temperature of 20 °C, and the distance Y between the wires is measured. The difference between Y and X is a measure of volume consistency. It meets the standard if it doesn’t exceed 10 mm [40].

Mass absorbability (n_w_) of mortars is calculated on the basis of the following Equation (1):(1)$$n_{w}=\frac{\left(m_{s}-m_{d}\right)}{m_{d}}*100\%$$
where:

m_s_–a mass of the water-saturated sample, g.

m_d_–a mass of the dry sample, g.

Tensile and compressive strength after 2 and 28 days of the mortar maturation was performed in compliance with standard EN 196-1 [17]. The samples were removed from forms after 48 h and kept in water until the analysis. The tests were executed on maturing samples in the temperature of 20 ± 2 °C. Furthermore, it was evaluated if the beams of mortars changed their volume to confirm or eliminate the swelling effect.

## 5. Results and Discussion

### 5.1. Physical and Chemical Properties of MSWI Slag

Table 5 presents the basic technical properties of tested slag from MSWI plant. The total moisture content of the slag amounts to 4.5%. This value is higher than recommended in the standard (≤ 1.0%). Higher moisture content may affect puzzolana activity. Moisture content in the slag may be lowered by extending the period of seasoning.

Bulk density of tested slag reached 1700 kg/m^3^, which is higher than the bulk density of the tested cement (900–1500 kg/m^3^).

Among physical properties which GBFS needs to meet to be acceptable for use as the addition of type II to the concrete, size of specific surface area was defined.

The size of the specific surface area of the tested slag was 3200.0 cm^2^/g, so the slag meets the requirements for GBFS according to the standard.

The undesirable components of the possible mineral additive include an excessive amount of compounds of sulfur, chloride and unburnt coal. The high content of the unburnt coal (C > 5%) may increase water demand and decrease frost resistance of mortars or concretes made with these mortars. The concentration of chloride, sulfur and organic coal in the tested slag was below 1.0%. Low content of the above elements (S, TOC, Cl) was reflected in a very low leachability.

Figure 4 presents the results of the loss on ignition (LOI) for the tested slag (S). LOI was determined by heating slag samples to the point of the solid mass in a muffle furnace at two temperatures: 60 °C and 950 °C in the oxidizing atmosphere. In the analyzed case, it was found that LOI in 600 °C meets the permissible criteria for the disposal at the landfill of wastes other than hazardous and neutral (LOI ≤ 8%). At the same time, loss on ignition for the slag was determined. The samples were tested by roasting in the temperature of 950 °C through the extended time of a maximum of 1 h. This parameter is essential for the mortar and concrete. The permissible limiting value of LOI for GBFS is ≤ 3%. Significant loss on ignition in ash may result in deterioration of the workability of mortar and concrete. In the tested slag, LOI reached 5.56%.

Table 6 presents leachability of the selected contaminants from the slag, which may constitute a nuisance to the environment or adversely affect mechanical properties of the concrete mixture which would influence the concrete durability.

Secondary wastes from MSWI may pose a problem for the environment due to the high content of chloride salts and sulphates. Similar observations can be found in the publication [42]. Leachability of sulphates from the tested slag does not exceed a permissible value for the disposal at landfills for neutral wastes. However, the leachability level of sulphates is too high, slightly exceeding the acceptable value for the neutral wastes. The chemical requirements limit the content of chloride ions in the tested slag to 0.1%. Moreover, the concentration of sulfur ions must be not higher than 2.5%.

The aqueous extracts from the tested slag have been prepared to measure heavy metals concentration. These included: barium, zinc, copper, lead, cadmium, chrome, cobalt, iron, manganese and nickel. (Table 7).

The leachability level of heavy metals from the tested slag was very low. Only Pb level exceeded about 0.1 mg/kg the permissible value for the waste stored in the landfills for the neutral waste. The concentration of the other metals (Zn, Cd, Ni) did not exceed the permissible values, and in many cases (Ba, Cu, Cr, Co, Fe, Mn) it was below the detection threshold.

Together with the physicochemical tests, mortar with 30% addition of slag from MSWI was designed and produced. The mortars were maturing for 28 days in the laboratory conditions. After that, aqueous extracts were prepared. The leachability of contaminants from the mortar with the addition of the slag could be affected by the form of the mortar (monolith or crushed) which subsequently may influence the environmental nuisance.

In case of the monolith form, surface release process and diffusion determine to a large extent, the leachability level. In case of crushed mortar, the leachability level depends on the percolation process. The article presents tests for the leachability of contaminants for both forms of the mortars.

The MSWI slag should be tested to check its chemical composition before each use. Slag MSWI was evaluated over a period of two years. The slag composition fluctuates mainly in the scope of CaO and Cr due to the technology used in incineration plants in its valorisation. As a response to comparison requirements in the field of chemical fabric composition, granulated blast furnace slag GBFS can be adopted EN 15167–1: 2007 standard: “Ground granulated blast furnace slag for use in concrete, mortar and grout-Part 1: Definitions, specifications and compliance criteria” [16] presents the chemical (Table 1) that must be met so that, for example, ground granulated blast furnace slag can be used as a type II additive in the composition of concrete. Legal requirements for MSWI slag are necessary so that it can be used as an additive to concrete. We can only refer to requirements for blast furnace slag GBFS. It is necessary due to the heterogeneity of the slag from the MSWI installation, and the significant content of harmful components (unburned coal and sulfur compounds) are the main causes of low concrete resistance. If they refer to the requirements for blast furnace slag, it should meet the chemical requirements given in Table 8, and such should also be used for MSWI slag. The comparison of data from Table 2, Table 3, Table 4, Table 5, Table 6, Table 7 and Table 8 suggest that MSWI slag can be used in building practice because the analyzed standard requirements for slag as a replacement of cement material are mostly filled.

Table 9 presents the properties of granulated blast furnace slag about the requirements of the EN 197-1 standard “Cement-Part 1. Composition, requirements and compliance criteria for common cement [18].” Table 2. Granulated blast furnace slag in the requirements of the EN 197-1 standard [18].

GBSF slag is a cement component with latent hydraulic properties, and its activity depends on the alkalinity of the slag and thus on the CaO content. Too much SiO_2_ reduces the hydraulic activity of the slag. In case of tested MSWI the proportion between CaO, MgO and SiO_2_ is adequate (Table 2).

### 5.2. The Research Results of Properties of Fresh and Hardened Mortar with MSWI Slag

#### 5.2.1. Properties of Fresh Mortar in the Aspect of Requirements of Slag as An Addition to Mortars

Test results indicated that the analyzed cement paste (Figure 5 and Figure 6) containing 30% of blast furnace slag maintains complete volume following the requirements of standard EN 197-1:2002 [18]. Le Chateliera’s test (Figure 5) proves that the permissible 10 mm is not exceeded. Change of the volume of the mortar was 0.1%.

Setting time of cement paste with 30% blast furnace slag (Figure 6) tested in automatic Vicatronic (Figure 7) was 325 min and ended after 555 min. For comparison, the binding of Portland cement (CEM II/B-S 42% N) starts after 240 min and ends after 300 min. The cement paste time for binders with 30% of blast furnace slag (S) meets the requirements of standard EN 197-1 [18] concerning the minimum start of the binding time, which is 60 min for CEM I.

Table 10 presents the properties of fresh mortars. The obtained results indicate that the fresh mortar with the 30% addition of the slag-MSWI (S-MSWI) has similar properties than the mortar, which contained 30% of GBSF. The flow diameter of the tested mortar mixtures (CEM I, CEM I + 30% S-MSWI, CEM I + 30% GBFS) in time is on a similar level.

Figure 8 presents measurements of consistency of mortar with 30% addition of blast furnace slag. The research results indicate that the flow diameter of the mortar with GBSF and MSWI slags is similar, and is about 16 cm. The time of keeping the consistency for 60 min is also similar.

Air content determined in the fresh mortar with MSWI and GBSF slag reached 2.5% and 3.1% respectively, which means that there is no significant influence of the type of slag on the air content of mortar.

Mentioned test results proved that tested cement pastes and mortars with 30% addition of MSWI slag meet the requirements of EN 196-3 [40].

#### 5.2.2. Properties of Hardened Mortar in the Aspect of Requirements of Slag as An Addition to Mortars

Figure 9 showed that the mortar mortars with the 30% addition of slag did not swell, and preserved their original size. No performances or structure defects were observed.

Figure 10, Figure 11 and Figure 12 present the results of compressive strength and tensile strength tests of mortars with the 30% addition of slag (S) from MSWI tested in automatic apparatus (Figure 8). For comparing the obtained results, the same tests were performed on the mortar with the 30% addition of the granulated blast furnace slag. Portland cement CEM I 42.5 R (CEM I) was used as a reference mortar.

Mortar with a binder containing 30% of blast furnace MSWI slag showed lower compressive strength in the initial period of hardening (after two days) when compared to the mortar with 30% of GBFS (Figure 8). However, after 28 days of maturation, the compressive strength of the tested mortars did not differ much (2 MPa). Based on the obtained results, it may be stated that the designed mortar matrixes may be used for similar engineering purposes.

The water absorption of the 28 days hardened mortar made with the 30% addition of the blast furnace slag reaches a level of 9%. For comparison, the absorbability of the mash with GBSF slag is 7%. In case of Portland cement (CEM I) water absorption falls in the range of 3%–8%. It may be stated that the addition of slag does not significantly affect the water absorption of concrete. It is known that after 90 days, the absorbability of mortar with slag changes, reducing its value [12].

Analysis of the mechanical properties of mortars after 2 and 28 days of hardening indicates that slag (S) from MSWI plant is a component that fills microstructure of mortars without pozzolan properties, unlike GBFS. The reason for this is the lack of the vitreous phase in case of MSWI slag.

In case of MSWI slag particularly undesirable are reactions between cement and aluminum and zinc found in the wastes. Concrete may swell because of the salts such as sulfurs (SO_4_), which can be found, for example, in the soil, in the direct contact with the concrete. It leads to a chemical reaction with aluminum contained in the cement generating an expansive substance. These phenomena may lead to inciting internal tensile forces and consequently to the occurrence of cracks, in case of excessive concrete resistance [43]. The reasons and the process of the above mentioned chemical reactions are described in details in the publications [44]. It is known that these reactions cause swelling and even cracking of concrete.

Furthermore, non-homogeneity of the slag from MSWI plant and significant content of harmful components (unburnt coal and sulfur compounds) are the main reasons for low resistance of the concrete and limit the way of its use. Depending on the composition, concrete strength reaches 2–10 MPa. Slag from MSWI plant shall not be used for the production of reinforced concretes and concretes exposed to permanent humidity over 75%. When the amount of the substance mentioned above is high, mortar/concrete may tend to change volume (shrinkage, swelling) under the wet conditions. The reason is usually the content of unburnt coal. Water absorption of the hearth slag concrete, depending on the type of slag and composition of concrete, equals 15%–25%. Frost resistance of hearth slag concrete is generally satisfactory, under the condition that the slag used does not contain a large amount of unburnt coal (C < 5%), and the concrete resistance is not lower than 5 MPa [12].

Concrete with the additive of slag from MSWI is used in the building industry almost exclusively for the production of various types of wall hollow bricks and less often for the construction of monolith walls at 1–2 floor in the individually constructed houses. Due to the unfavorable results of experiments, it is no longer used in the monolith wall, and only small wall elements are made of it.

To be suitable for use, the slag from MSWI plant in a responsible way, it needs to be seasoned and processed. Depending on the original composition and eventual destination, processing of slag may assume various forms (seasons, rinsing, washing, exposing to sodium hydroxide, removal of heavy metals, vitrification) to assure the expected quality of the concrete [8]. One of the most frequent problems will be the necessity to reduce aluminum and the amount of glass contained in the slag and control, among others the amount of unburnt coal, chloride, zinc and sulfur. Also, the possibility that the possible hazardous substances get through to the environment needs to be limited. [43,44].

Slags from MSWI require constant analysis, mainly referring to the chemical compositions. Based on the reports, a correct decision could be taken concerning their possible use, for example, for the production of concretes. A search for concrete with the addition of slag from MSWI plant that is environmentally friendly and that has excellent mechanical properties, suitable volume, and low leachability of contaminants is still ongoing [13,43,44,45].

#### 5.2.3. The Influence of Mortar with MSWI on Environment Research Results

Table 11 presents the leachability of the selected contaminants from the 28 days hardened mortars. High pH value (pH > 11) may indicate high mobilization of heavy metals, chloride salts and sulphates. The leachability of standard contaminants (chlorides, sulphates, TOC) does not exceed permissible levels for depositing wastes in the landfills for neutral wastes.

The leachability of heavy metals from monolithic concrete forms after 28 days of their maturation is presented in Table 12.

The obtained levels of concentration of heavy metals in the aqueous extract were compared to the permissible values for wastes designated to the disposal at the landfills other than hazardous and neutral [41]. It needs to be noted that most of the values of heavy metals leachable from the mortar with the 30% addition of slag were definitely below the acceptable limit. The only value which exceeded the limit was Ni level. The concentration of these elements exceeded twice the requirements for the disposal at the neutral landfills.

Within the course of the conducted tests, also crushed mortar was exposed to the activity of the elusion liquid. Table 13 shows the concentration of the selected contaminants, which may be hazardous for the environment, in the aqueous extracts from the crushed mortar after 28 days of maturation. The increase in pH was noted (pH > 12). Leachability of chloride salts in the crushed mortar increased more than 20 times in respect of the monolith mortar, exceeding the acceptable level for disposal at the landfill for the neutral wastes. Leachability of contaminants from the crushed mortars with the 30% addition of slag from MSWI did not exceed the permissible values for the wasted destined for disposal at landfills for wastes other than hazardous and neutral.

Table 14 presents the content of heavy metals in the aqueous extracts from crushed mortar after 28 days of maturation. It should be underlined that none of the concentration values of the heavy metals leachable from the crushed mortar made with 30% addition of slag exceeds the permissible value for the depositing at landfills other than hazardous and neutral. Only, similarly to the mortar, the content of Ni exceeds the limit. Fair value of this element exceeded the requirements for the neutral landfills more than four times.

## 6. Conclusions

The obtained results give a chance for environmentally friendly use of the slag in the construction industry. In many cases, the analyzed matter may constitute the best and the cheapest long term solution in the waste management economy, filling up the gap in the market in respect of the diminishing anthropogenic resources.

It is indispensable to legally determine the requirements for the slags from MSWI plants so that they could be used as an additive to the mortar and also in concrete. Mechanical tests showed that the MSWI slag might constitute the partial replacement of cement in mortar and concrete; the requirements of EN 196-3 [40] were fulfilled. Slag from MSWI plants assures similar aeration and consistency of the fresh mortar, also within time, similar to granulated blast furnace slag. Change of the volume of the standard slurry with 30% of the slag from MSWI equals 0.1%, which meets the requirements for the cement with a large margin. Setting time of the cement paste with MSWI slag is 125 min longer than the cement with blast furnace slag. Mechanical parameters of mortars with MSWI slag after 2 and 28 days of maturation are lower than the mortar with the milled blast furnace slag, but cement meets the requirements for CEM II/B-S 32 N in this respect. Water absorption of mortars with MSWI slag is 3.0% higher than mortars with milled granulated blast furnace slag. However, this value may be limited by lowering the amount of water added to the cement in the mortar.

Conducted chemical tests proved that the use of MSWI slag in mortar would not cause any environmental nuisance. Furthermore, many physical and chemical properties of the slag from MSWI are similar to widely used in mortars crushed, granulated blast furnace slag. It must be remembered that the composition of MSWI slag depends on the structure of the municipal wastes directed to the incineration plant. That is why further tests concerning the variability of the chemical content of the slag over time need to be conducted.

## Figures and Tables

**Figure 1 materials-13-01593-f001:**
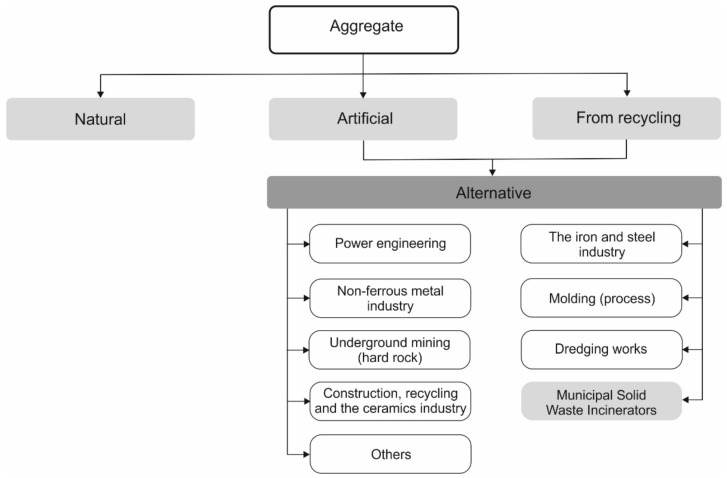
The direction of using anthropogenic raw materials in the circular economy (author: Czop, based on [5]).

**Figure 2 materials-13-01593-f002:**
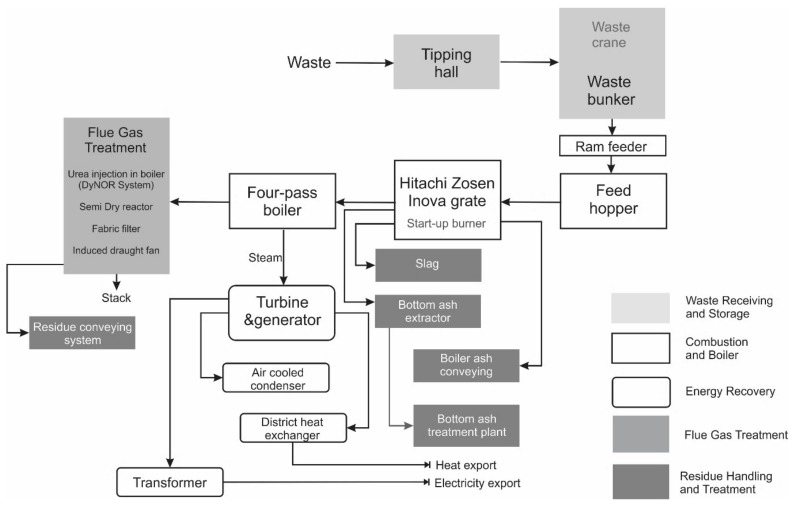
Simplified scheme of an analyzed municipal solid waste (MSW) incinerator (author: M. Czop based on [2]).

**Figure 3 materials-13-01593-f003:**
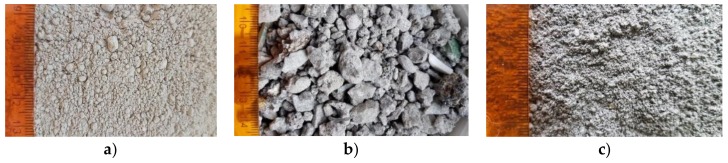
Analyzed materials: **a**) cement (CEM I), **b**) municipal solid waste incineration (MSWI) raw slag and **c**) ground slag (authors’ photo).

**Figure 4 materials-13-01593-f004:**
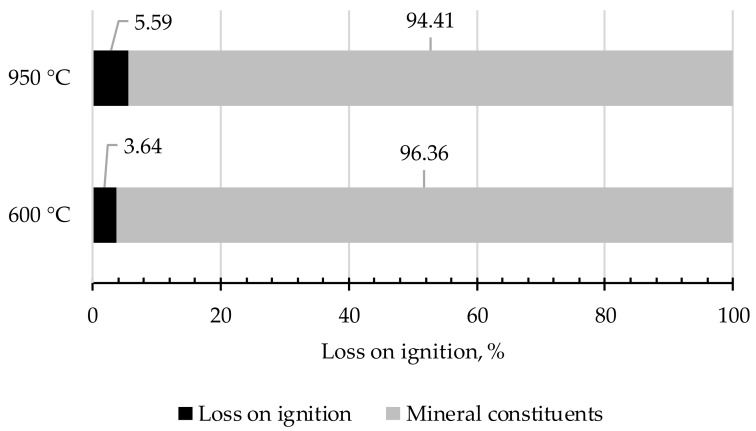
Loss on ignition (LOI) in the tested slag from MSWI.

**Figure 5 materials-13-01593-f005:**
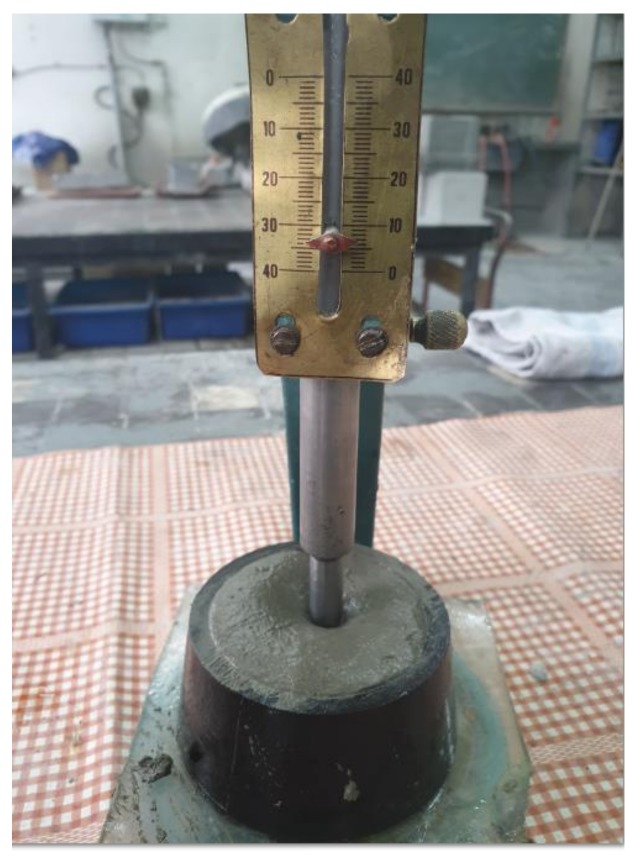
The view oement paste with MSWI slag under Chateliera’s test (authors’ photo).

**Figure 6 materials-13-01593-f006:**
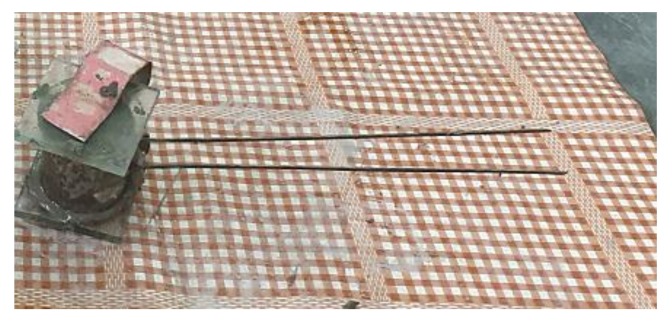
The view of normalized cement paste with MSWI slag in Vicat’s apparatus (authors’ photo).

**Figure 7 materials-13-01593-f007:**
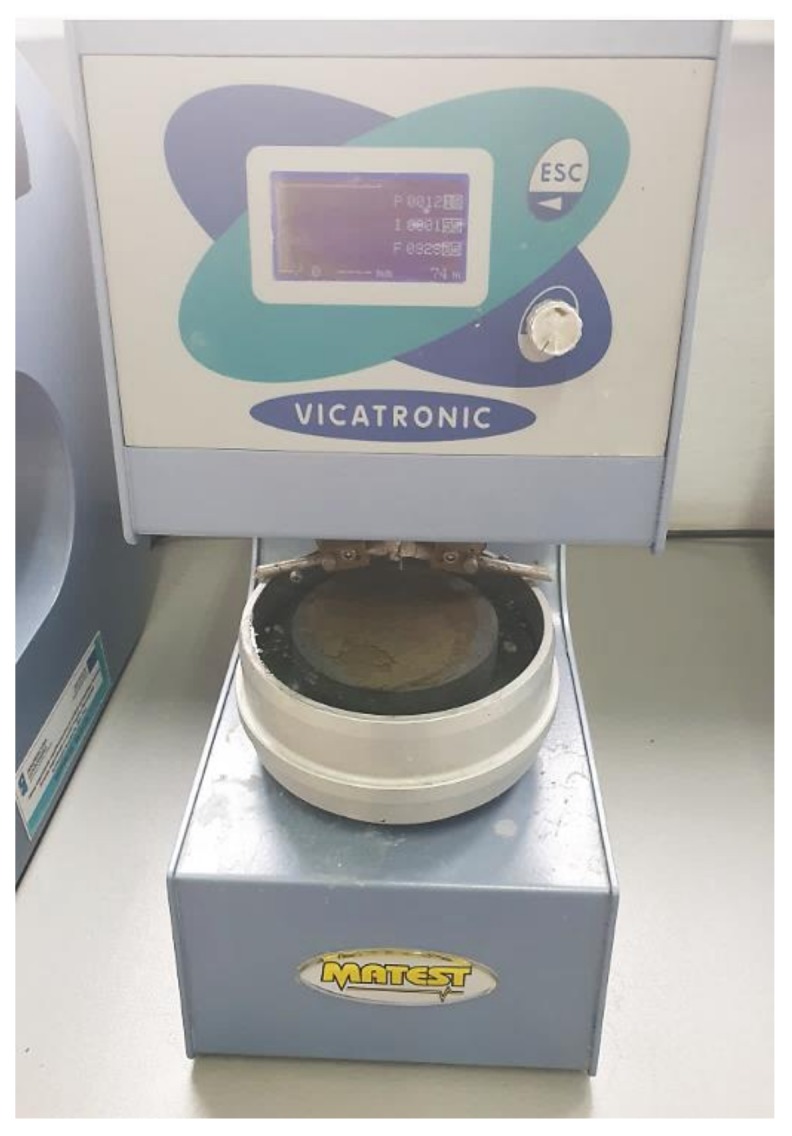
The view of cement paste with MSWI slag in automatic Vicastronic under measuring setting time (authors’ photo). No volume change of cement paste was observed under measuring of setting time, which could affect the correct measurement.

**Figure 8 materials-13-01593-f008:**
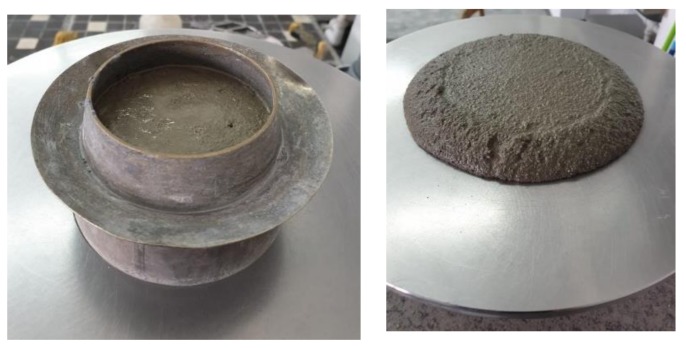
The flow of fresh mortar with the addition of 30% of MSWI slag after shaking 15 times (authors’ photo).

**Figure 9 materials-13-01593-f009:**
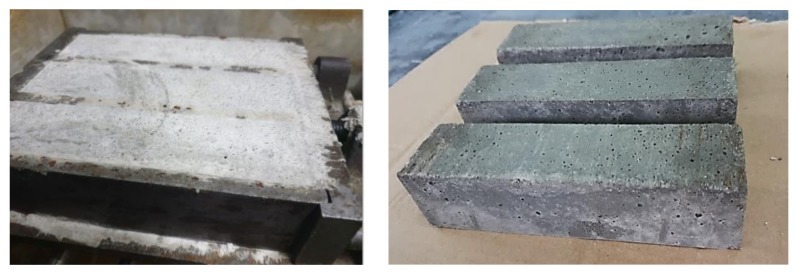
The view of mortars incorporating 30% of slag from MSWI in form and samples were taken from after 1 day with 40 × 40 × 160 mm (authors’ photo). No change of the mortar volume was observed.

**Figure 10 materials-13-01593-f010:**
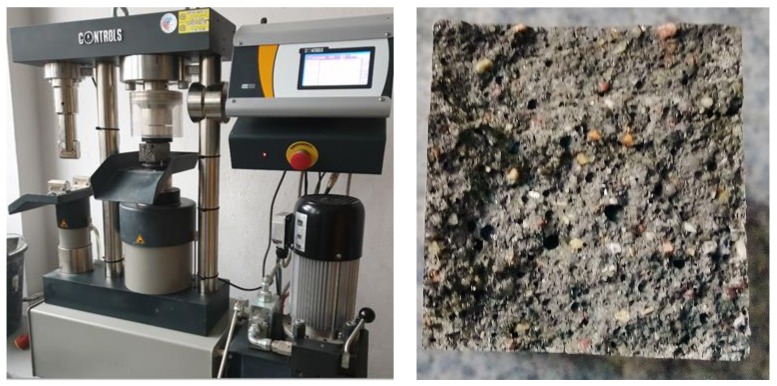
The view of compressive strength and tensile strength testing of mortars in the automatic apparatus and sample fracture structure (authors’ photo).

**Figure 11 materials-13-01593-f011:**
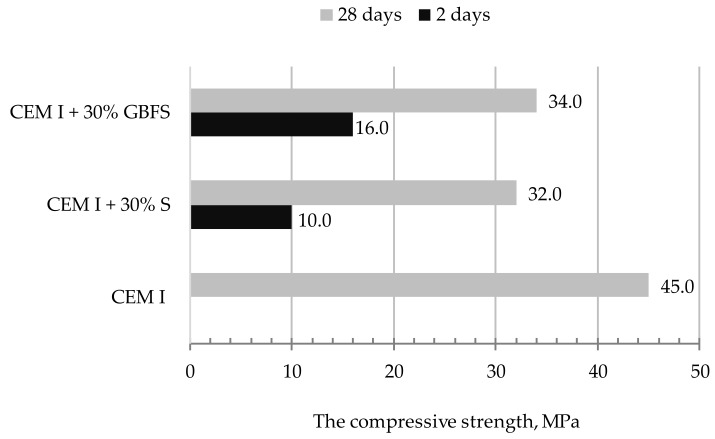
Results of compressive strength of mortars after 2 and 28 days of maturing.

**Figure 12 materials-13-01593-f012:**
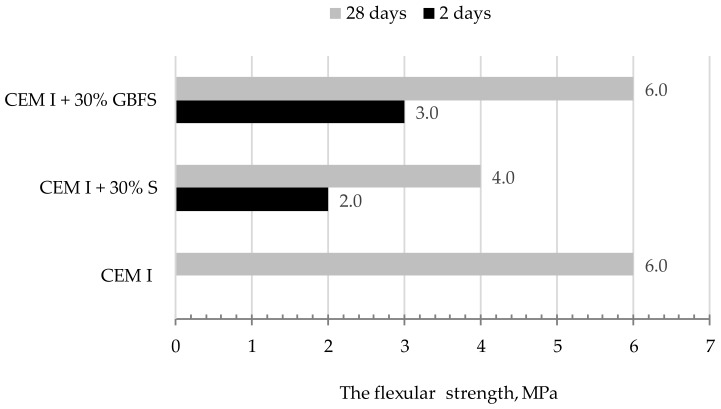
Results of bending tensile strength tests of mortars after 2 and 28 days of maturing.

**Table 1 materials-13-01593-t001:** The requirements for the milled granulated blast furnace slag [16].

Parameter	Symbol	Unit	Standard Requirement [16]
Specific surface area	−	cm^2^/g	≥ 2750.0
Magnesium oxide	MgO	%	18.0
Sulfide	S^2−^		≤ 2.0
Vitreous phase	−		≥ 67.0
Sulfates	SO_3_		≤ 2.5
Loss on ignition	LOI		≤ 3.0
Chloride	Cl^−^		≤ 0.1
Moisture	M_T_		≤ 1.0

**Table 2 materials-13-01593-t002:** Content of oxides (%) in the tested materials.

Properties	Symbol	CEM I	Slag-MSWI
Silicon dioxide	SiO_2_	14.00	57.90
Iron(III) oxide	Fe_2_O_3_	3.03	4.97
Aluminum oxide	Al_2_O_3_	7.47	10.80
Manganese(II,III) oxide	Mn_3_O_4_	0.12	0.12
Titanium dioxide	TiO_2_	1.09	0.50
Calcium oxide	CaO	51.20	12.50
Magnesium oxide	MgO	1.61	1.73
Sulfur trioxide	SO_3_	9.59	0.74
Phosphorus pentoxide	P_2_O_5_	1.01	0.74
Sodium oxide	Na_2_O	3.05	6.61
Potassium oxide	K_2_O	3.77	0.95
Barium oxide	BaO	0.14	0.14
Strontium oxide	SrO	0.05	0.06

**Table 3 materials-13-01593-t003:** Composition mix of mortars, expressed in gram.

Type of Waste	Symbol of Mortar	CEM I, g	Water, g	Standard Sand, g [17]
Reference sample from Portland cement 42.5 R	CEM I	450	225	1350
Cement + 30% slag	CEM I + 30% S	315	135	1350
Cement + 30% Granulated Blast Furnace Slag	CEM I + 30% GBFS	315	135	1350

**Table 4 materials-13-01593-t004:** Heavy metal concentration, expressed in mg/kg.

Properties	Symbol	CEM I	Slag-MSWI
Zinc	Zn	617.00	1621.00
Copper	Cu	94.70	1918.00
Lead	Pb	87.80	687.00
Nickel	Ni	20.80	81.00
Chrome	Cr	113.00	342.00
Cadmium	Cd	3.30	3.35
Arsenic	As	6.11	16.50
Vanadium	V	34.40	30.00
Thallium	Tl	< 1.00	< 1.00
Mercury	Hg	0.07	0.24

**Table 5 materials-13-01593-t005:** Basic technical properties of tested slag.

Properties	Symbol	Unit	Slag
Moisture	M	%	4.48
Bulk density	ρ_b_	kg/m^3^	1700.0
Specific surface area	S	cm^2^/g	3200.0
Total carbon	C	%	2.26
Total organic carbon	TOC	%	0.52
Sulfur	S	%	0.78
Chlorine	Cl	%	0.12

**Table 6 materials-13-01593-t006:** Leachability of selected contaminants of tested slag, expressed in mg/kg (with pH exception).

Properties	Symbol	Slag-MSWI	Criteria for Landfills for [41]	
			Inert Waste	Non-Hazardous Waste
pH	pH	7.9	−	minimum 6
Total Carbon	TC	118.00	−	−
Total Organic Carbon	TOC	BLQ*	30000	−
Total Inorganic Carbon	TIC	BLQ*	−	−
Chloride	Cl^−^	780.00	800	15000
Sulphate	SO_4_^2−^	1157.41	1000	20000
Phosphate trianion	PO_4_^−^	100.00	−	−
Potassium	K	354.60	−	−
Calcium	Ca	878.40	−	−
Lithium	Li	2.70	−	−
Sodium	Na	1104.00	−	−
The sum of chloride and sulphate	TDS	1937.41	4000	60000

BLQ*–Values below the limit of quantification.

**Table 7 materials-13-01593-t007:** Content of heavy metals in water extracts from the slag tested, expressed in mg/kg.

Properties	Symbol	CEM I	Slag-MSWI	Criteria for Landfills for [41]	
				Inert Waste	Non-Hazardous Waste
Bar	Ba	BLQ*	BLQ*	20	100
Zinc	Zn	BLQ*	0.14	4	50
Copper	Cu	0.64	BLQ*	2	50
Lead	Pb	0.15	0.60	0.5	10
Cadmium	Cd	BLQ*	0.04	0.04	1
Chrome	Cr	50.1	BLQ*	0.5	10
Cobalt	Co	BLQ*	BLQ*	−	−
Iron	Fe	1.80	BLQ*	−	−
Manganese	Mn	BLQ*	BLQ*	−	−
Nickel	Ni	2.08	0.22	0.4	10

BLQ*–Values below the limit of quantification.

**Table 8 materials-13-01593-t008:** The chemical requirements to be met by the slag GBFS and in comparison the research results of MSWI.

Ingredient Content	The Standard Specification for Content (% by Weight) in Case of GBFS
Magnesium oxide (MgO)	≤ 18.0
Sulfides (S^2−^)	≤ 2.0
Vitreous phase	≥ 67.4
Sulfates (SO_3_)	≤ 2.5
Roasting loss	≤ 3.0
Chlorides (Cl^−^)	≤ 1.0
Humidity	≤ 1.0

**Table 9 materials-13-01593-t009:** The requirements of granulated blast furnace slag.

Requirements of EN 197-1	Value
content of the vitreous phase	> 95.0%
CaO + MgO + SiO_2_	≥ 2/3
Ca + MgO/ SiO_2_	≥ 1.0%

**Table 10 materials-13-01593-t010:** Properties of fresh mortar.

Properties	Unit	CEM I	CEM I + 30% S-MSWI	CEM I + 30% GBFS
Flow value (initial)	mm	150.0	160.0	160.0
Flow value (after 60 min)	mm	140.0	150.0	150.0
Air content	%	2.5	3.1	2.9

**Table 11 materials-13-01593-t011:** Leachability of selected contaminants from mortar expressed in mg/kg (with pH exception).

Properties	Symbol	CEM I	CEM I + 30% S	Criteria for Landfills for [41]	
				Inert Waste	Non-Hazardous Waste
pH	pH	8.9	11.1	−	min. 6
Total Carbon	TC	89.20	89.20	−	−
Total Organic Carbon	TOC	55.80	55.80	30,000	−
Total Inorganic Carbon	TIC	33.40	33.40	−	−
Chloride	Cl^−^	161.28	46.08	800	15,000
Sulphate	SO_4_^2−^	633.56	292.09	1000	20,000
Phosphate trianion	PO_4_^−^	20.33	24.00	−	−
Potassium	K	3.56	3.00	−	−
Calcium	Ca	4.97	2.54	−	−
Lithium	Li	BLQ*	BLQ*	−	−
Sodium	Na	3.08	2.08	−	-
The sum of chlorides and sulphates	TDS	794.84	338.17	4000	60,000

BLQ*–Values below the limit of quantification.

**Table 12 materials-13-01593-t012:** Leachability of heavy metals from mortar (integral), expressed in mg/kg.

Properties	Symbol	CEM I	CEM I + 30% S	Criteria for Landfills for [41]	
				Inert Waste	Non-Hazardous Waste
Bar	Ba	BLQ*	BLQ*	20	100
Zinc	Zn	BLQ*	BLQ*	4	50
Copper	Cu	BLQ*	BLQ*	2	50
Lead	Pb	0.11	0.22	0.5	10
Cadmium	Cd	0.01	0.04	0.04	1
Chrome	Cr	BLQ*	BLQ*	0.5	10
Cobalt	Co	BLQ*	BLQ*	−	−
Iron	Fe	BLQ*	BLQ*	−	−
Manganese	Mn	0.04	BLQ*	−	−
Nickel	Ni	BLQ*	0.84	0.4	10

BLQ*–Values below the limit of quantification.

**Table 13 materials-13-01593-t013:** Leachability of selected contaminants from crushed mortar, expressed in mg/kg (with pH exception).

Properties	Symbol	CEM I	CEM I + 30% Slag-MSWI	Criteria for Landfills for [41]	
				Inert Waste	Non-Hazardous Waste
pH	pH	12.4	12.8	−	min. 6
Total Carbon	TC	68.20	106.20	−	−
Total Organic Carbon	TOC	54.90	66.40	30,000	−
Total Inorganic Carbon	TIC	13.30	39.80	−	−
Chlorides	Cl^−^	1198.08	944.64	800	15,000
Sulfate	SO_4_^2−^	283.00	373.00	1000	20,000
Phosphate trianion	PO_4_^−^	25.33	12.33	−	−
Potassium	K	299.13	479.67	−	−
Calcium	Ca	590.43	1335.33	−	−
Lithium	Li	2.77	4.90	−	−
Sodium	Na	94.43	182.40	−	−
The sum of chlorides and sulphates	TDS	1200.91	948.37	4000	60,000

**Table 14 materials-13-01593-t014:** Leachability of heavy metals from crushed mortar, expressed in mg/kg.

Properties	Symbol	CEM I	CEM I + 30% S	Criteria for Landfills for [41]	
				Inert Waste	Non-Hazardous Waste
Bar	Ba	BLQ*	BLQ*	20	100
Zinc	Zn	BLQ*	BLQ*	4	50
Copper	Cu	0.07	BLQ*	2	50
Lead	Pb	0.55	BLQ*	0.5	10
Cadmium	Cd	BLQ*	BLQ*	0.04	1
Chrome	Cr	BLQ*	BLQ*	0.5	10
Cobalt	Co	BLQ*	BLQ*	−	−
Iron	Fe	BLQ*	BLQ*	−	−
Manganese	Mn	BLQ*	BLQ*	−	−
Nickel	Ni	0.13	1.87	0.4	10

BLQ*–Values below the limit of quantification.

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
