# Peer review of "Use of Slag from the Combustion of Solid Municipal Waste as A Partial Replacement of Cement in Mortar and Concrete"

_materials, 2020, doi:10.3390/ma13071593_

Round 1
Reviewer 1 Report
This is an interesting paper testing the potential applicability of MSWI ashes in cement. Such practice has been widely applied for other industrial by-products (e.g. steel making slags) but has yet to be tested at scale for MSWI ashes to my knowledge. As such there is novelty associated with the work. One of the key findings relates to issues of leachate generation and salt content diminishing structural performance. These are interesting findings. The authors could improve some of the context for the work with clearer reference to some of the environmental issues associated with reuse of alkaline residues. It would also be very good to see clarity on sample numbers (initial sample size and replicates used in the tests). Heterogeneity is a key issue with MSWI ashes (as the authors conclude) so demonstrating how the authors approached this problem with sample design would be useful to see. Some other suggestions are made to help improve clarity of the work. Otherwise, this is a novel contribution and makes a timely contribution to the literature on waste re-use and circular resource use for the growing EfW sector.
Specific comments:
Line 36-37 - reference needed for the 1/10th EU waste arisings figure. This is a good point to make.
Line 51 - some broader context would be useful here on the push for Energy from Waste schemes. This has been a clear pattern across the EU and helps highlight the novelty of the work. Most combustion ashes end up in landfill so finding alternative end uses is valuable for CE efforts.
- Line 75 - were not - avoid contraction of words with apostrophes.
- Line 84 - please provide more detail on the potential negative impacts. A key one is obviously leaching of potentially harmful oxyanion-forming elements under high pH (e.g. Cornelis et al., 2008; Gomes et al., 2016).
- Table 3 - explain sample codes in the legend also for the cement and slag.
- Table 3 - Chromium rather than chrome (and in other tables)
Line 256 - pozzolanic?
- Table 5 - include sample size and range for the data presented.
- Line 266 - it would be informative to see references here on the increased moisture reducing front resistance.
- Line 308 - add references here to the alkaline leaching risk (as above - Cornelis and Gomes)
- Table 9 - Barium not Bar
References:
Cornelis, G., Johnson, C.A., Van Gerven, T. and Vandecasteele, C., 2008. Leaching mechanisms of oxyanionic metalloid and metal species in alkaline solid wastes: a review. Applied Geochemistry, 23(5), pp.955-976.
Gomes, H.I., Mayes, W.M., Rogerson, M., Stewart, D.I. and Burke, I.T., 2016. Alkaline residues and the environment: a review of impacts, management practices and opportunities. Journal of Cleaner Production, 112, pp.3571-3582.
Author Response
Thank you very much for your constructive reviews. Your comments are very valuable to us. We will use your experience in our further scientific work. We thank you for your time.
Answers to reviews 1. Line 36-37 - reference needed for the 1/10th EU waste arisings figure. This is a good point to
make.
The sentence has been corrected and made more specific: In 2018, 2.5 billion Mg of waste was
generated in the EU countries. Municipal waste accounts for about 10% of the total stream. 2. Line 51 - some broader context would be useful here on the push for Energy from Waste
schemes. This has been a clear pattern across the EU and helps highlight the novelty of the
work. Most combustion ashes end up in landfill so finding alternative end uses is valuable for
CE efforts.
Energetic use of waste is medicine for a low-quality fraction. These wastes, due to their high
heterogeneity and amount of impurities, cannot be sent for recycling (material, raw material or
chemical). Until now, they ended their life cycle in the landfill. This solution is a waste of resources.
Energetic use brings great ecological benefits, which are associated with reducing the amount of waste
deposited. In addition to ecology, economics is important. Economic benefits are associated with
saving exhaustible resources of fossil fuels.
3. Line 75 - were not - avoid contraction of words with apostrophes.
This has been corrected.
4. Line 84 - please provide more detail on the potential negative impacts. A key one is obviously
leaching of potentially harmful oxyanion-forming elements under high pH (e.g. Cornelis et al.,
2008; Gomes et al., 2016).
A potential threat when using secondary waste from the thermal degradation of municipal waste for
concrete production is the migration of heavy metals to the soil and water environment. High levels of
chloride salts and sulfates may prove to be environmentally problematic. The negative effect of salt
migration into the environment may be the extinction of living organisms. And this can lead to a
reduction in species biodiversity. High salt contents can inhibit plant growth, reduce their size and the
number of leaves and roots. As a result of the accumulation of salt in the soil, nutrients such as
phosphorus, calcium and potassium may be leached.
5. Table 3 - explain sample codes in the legend also for the cement and slag.
Table 3 explains the abbreviations used.
6. Table 3 - Chromium rather than chrome (and in other tables)
Chrome replaced as suggested on Chromium
7. Line 256 - pozzolanic?
Thank you for the correction, an error has crept in. The word has been corrected.
8. Table 5 - include sample size and range for the data presented.
Table 5 presents basic information about the tested slag.
9. Line 266 - it would be informative to see references here on the increased moisture reducing
front resistance.
Together with an industrial partner, we are working on reducing the amount of water in waste. Tests
are currently underway, and we hope that we will soon be able to present the next publications and
the results of cooperation on this issue. Before applying slag to concrete and concrete, it must be dried
to 1% water.
10. Line 308 - add references here to the alkaline leaching risk (as above - Cornelis and Gomes)
The authors observed similar results in the publication [48]. 11. Table 9 - Barium not Bar
Bar replaced as suggested on Barium.
Reviewer 2 Report
Dear Dr. Monika Czop
The article is focused on the utilization of waste from MSWI. Landfilling is not an optimal solution. Therefore, ways are being searched for how to efficiently and economically manage these residues. The authors focused on the utilization of slag from the incineration of MSWI and its partial replacement in cement.The authors focused on characterization of raw materials, but mainly on leachability, determination of strength and determination of flexural strength.
According to my research, this topic is not new, but there are not many authors to pay attention to it and but the issue is topical in Poland.
- I appreciate the comparison (reference sample with GBFS). Because at the moment there is no reference sample for this kind of slag.
- I positively evaluate the procedure of analyzes, especially according to standards GBFS that have meaningful data and are a good source for further research, but also possible implementation in practice.
- I also positively evaluate the authors' own images, which give a graphical and clear overview of both the secondary sources and their distribution, and the place of slag formation.
- I appreciate a suitably chosen topic with regard to the environment, waste elimination, but also application possibilities.
Comments and questions:
Q1: In table 2 is the content of oxides% in the tested material. Tested CEM I does not comply with the standard composition according to research and cement prroduction. Standard if OPC, (CEM I 42,5R)the content of CaO, SiO2, Fe2O3 and Al2O3 (portland clinker) is above 92%. Which is not fulfilled in this case (the content in experiment is 75.7%), but the main component of CaO is minority 14% and SiO2 is the majority with the content of 51.20%. I propose a verified chemical composition or added the XRD record of these CEM I. In this state, this could significantly reduce the quality of the topic (research).
Q2: How often is tested slag for variability in chemical composition?
“In addition, the inhomogeneity of slag from the MSWI plant and the significant content of harmful constituents (unburned coal and sulfur compounds) are the main reasons for the low resistance of the concrete and limit the way it is used. "It is necessary to legally set requirements for MSWI slag in order to be used as an additive in concrete." Here we come across the chemical composition of this slag.” Link to citations in article.
Author Response
Thank you very much for your constructive reviews. Your comments are very valuable to us. We will use your experience in our further scientific work. We thank you for your time.
Answers to reviews
- In table 2 is the content of oxides% in the tested material. Tested CEM I does not comply with the standard composition according to research and cement prroduction. Standard if OPC, (CEM I 42,5R)the content of CaO, SiO2, Fe2O3 and Al2O3 (portland clinker) is above 92%. Which is not fulfilled in this case (the content in experiment is 75.7%), but the main component of CaO is minority 14% and SiO2 is the majority with the content of 51.20%. I propose a verified chemical composition or added the XRD record of these CEM I. In this state, this could significantly reduce the quality of the topic (research).
Thank you for your valuable comments.
Indeed, cement does not meet the requirements raised by the reviewer. We added this remark to the article. The tested cement was commercially available, and we did not influence its composition. The results were a surprise to us. We wrote about this in the article.
Unfortunately, we do not have a quick opportunity to examine XRD right now. Research is difficult due to limited access to the laboratory due to a pandemic.
- How often is tested slag for variability in chemical composition?
“In addition, the inhomogeneity of slag from the MSWI plant and the significant content of harmful constituents (unburned coal and sulfur compounds) are the main reasons for the low resistance of the concrete and limit the way it is used. "It is necessary to legally set requirements for MSWI slag in order to be used as an additive in concrete." Here we come across the chemical composition of this slag.” Link to citations in article.
PN-EN 15167–1: 2007 standard: "Ground granulated blast furnace slag for use in concrete, mortar and grout - Part 1: Definitions, specifications and compliance criteria" [21] presents both chemical (Table 1) and physical (Table 2) that must be met so that, for example, ground granulated blast furnace slag can be used as a type II additive in the composition of concrete. Legal requirements for MSWI slag are necessary so that it can be used as an additive to concrete. We can only refer to requirements for blast furnace slag. It is necessary due to the heterogeneity of the slag from the MSWI installation and the significant content of harmful components (unburned coal and sulfur compounds) are the main causes of low concrete resistance. If they refer to the requirements for blast furnace slag, it should meet the requirements given in Tables 1 and 2. And such should also be used for MSWI slag.
Table 1. Chemical requirements for the milled granulated blast furnace slag
|
Parameter |
Symbol |
Unit |
Standard requirement |
|
Magnesium oxide |
MgO |
% |
≤18.0 |
|
Sulfide |
S2– |
≤ 2.0 |
|
|
Vitreous phase |
- |
≥ 67.0 |
|
|
Sulfates |
SO3 |
≤ 2.5 |
|
|
Loss on ignition |
LOI |
≤ 3.0 |
|
|
Chloride |
Cl– |
≤ 0.1 |
|
|
Moisture |
MT |
≤ 1.0 |
Table 2. Physical requirementsor the milled granulated blast furnace slag
|
Parameter |
Unit |
Standard requirement |
|
Specific surface area |
cm2/g |
≥ 2750.0 |
|
7-day activity indicator |
% |
≥45.0 |
|
28-day activity indicator |
% |
≥70.0 |
|
Start of setting time |
min |
not more than twice the setting time for comparative cement |

Reviewer 3 Report
The research scope in interesting and the testing was adequate. The paper may be of interest for publication, given the raising environmental concern and the need for reducing the cement consumption at the same time as the waste management is improved. However, the paper needs to be greatly revised prior to publication. In particular, the materials and methods section must be rearranged to make the paper easier to follow. Also, the discussion section includes some elements that can be located in other parts of the paper.
The following lines include the main elements that should be amended, with identification of the lines.
Title. The title should not be misleading. Even though it can be understood in its present form, I believe that the indication of “Partial replacement of cement” would be appropriate.
Abstract. The abstract properly states the methodology and goals of the research.
Line 19. “For the use” could be replaced by “For its use”. GBFS should not be introduced without explanation the first time it appears in the text (in the abstract).
Introduction.
Line 36-37. The sentence is unclear. Please rewrite.
Lines 58-60. Please consider altering the punctuation of the sentence, as seems quite unnatural in English.
Figure 1. The figure caption does not properly address the figure’s content. Should “Aggregate” in the figure be interpreted as aggregates for concrete? If so, please explain in Figure caption. Also, it would be interesting to add a sentence in the main text that directs the attention towards Figure 1.
Line 75. “We only base on the experience”. Please rewrite from the Poland’s experience, within an academic register. The same can be applied to line 79.
Line 86. Please provide an alternative word for “parameters”.
Lines 88-90. Please provide reference for the data.
Line 91. As in line 75, the replacement of “we are missing standard” by “there is a lack of Polish/national standards” is encouraged.
Line 100. “To increase reliability” is hard to understand in this context. Please consider rewriting.
Line 106. Please amend “Residua waste”.
Figure 2. Please include some sentence such as “a scheme depicting the process of solid waste management is presented in Figure 2”.
Materials and methods.
Line 155. The beams should not be introduced as “materials”. It would be better to indicate that, for the tests, concrete beams were casted containing CEM I with a partial substitution of 30% slag. In these lines, the existence of the control (reference) samples without slag should be introduced.
Please provide data of “standard sand”.
Line 163. Please provide another word for “realized”.
Line 164. Up to that moment, slag has never been referred to as “gravel”. Please justify or rewrite.
Lines 170-171. Cast of the 40x40x160 mm beams.
Lines 176-177. This step should be rewritten. A possibility is to state it as “Tensile and compressive strength evaluation through destructive testing of the beams. The size of the beams should be introduced when they are cast, not later on in the text. Also, the test methods should be clearly stated regarding compressive and tensile testing. Was compressive and tensile strength estimated from beam tests? If so, please explain.
Lines 179-190. Is independent that testing from the beams? If so, please introduce it in the materials section as part of the characterization of the slag.
Lines 191-193. When was that testing conducted? Please organize information so it can be easily followed as a standard procedure.
Lines 214-220. This is a restatement of what should have been introduced in Lines 170-171. Please check coherence and discourse of all this section. Arguably, some information in lines 170-193 is redundant or should be reorganized.
Results and discussion.
Line 263. Please check slug/slag. In “so the size recommended …” please rewrite. I suggest it could be “so the slag meets the requirements for GBFS according to the standard…”.
Line 326. Please rewrite. Does crushed concrete refer to portions of the specimens after mechanical characterization? Please, specify the age of maturation in advance, not in line 336.
In order to make the paper is easier to follow, please consider reordering the results so that the fresh properties are discussed before the crushed properties, which were tested after mechanical characterization at 28 days.
Line 390. Please rewrite the sentence for more conventional syntactic order.
Lines 390-424. These statements should be presented in the introduction, to justify the research. If some of this content is included in the discussion section, it must clearly compare the general results with the results from the conducted tests.
Conclusions.
Line 427. Please replace “my” by “may”.
Line 438. Why previous?.
Author Response
Thank you very much for your constructive reviews. Your comments are very valuable to us. We will use your experience in our further scientific work. We thank you for your time.
Answers to reviews
Title. The title should not be misleading. Even though it can be understood in its present form, I believe that the indication of “Partial replacement of cement” would be appropriate.
As recommended by the Reviewer, the topic has been made more specific, and now it reads:
Use of slag from the combustion of solid municipal waste as a partial replacement of cement in mortar and concte
Abstract. The abstract properly states the methodology and goals of the research.
Line 19. “For the use” could be replaced by “For its use”. GBFS should not be introduced without explanation the first time it appears in the text (in the abstract).
We agree with the Reviewer's comments, "For the use" was changed to "For its use". The abbreviation the milled granulated blast furnace slag (GBFS) is explained.
Introduction.
- Line 36-37. The sentence is unclear. Please rewrite.
The sentence has been corrected and made more specific: In 2018, 2.5 billion Mg of waste was generated in the EU countries. Municipal waste accounts for about 10% of the total stream.
- Lines 58-60. Please consider altering the punctuation of the sentence, as seems quite unnatural in English.
The whole article has been subjected to language correction again, and the sentence has been changed.
- Figure 1. The figure caption does not properly address the figure’s content. Should “Aggregate” in the figure be interpreted as aggregates for concrete? If so, please explain in Figure caption. Also, it would be interesting to add a sentence in the main text that directs the attention towards Figure 1.
Figure 1 presents the general assumption that Circular Economy promotes. One of the assumptions is the recycling of waste resulting from the thermal degradation of municipal waste. We wanted to present the general assumption of Circular Economy and to indicate that the tested slag falls within the promoted idea.
- Line 75. “We only base on the experience”. Please rewrite from the Poland’s experience, within an academic register. The same can be applied to line 79.
As suggested by the Reviewer, it was entered: Poland’s experience, within an academic register
- Line 86. Please provide an alternative word for “parameters”.
The word parameters has been replaced by the word properties.
- Lines 88-90. Please provide reference for the data.
The literature has been supplemented.
- Line 91. As in line 75, the replacement of “we are missing standard” by “there is a lack of Polish/national standards” is encouraged.
According to the Reviewer's comments, it was changed into there is a lack of national standards
- Line 100. “To increase reliability” is hard to understand in this context. Please consider rewriting.
Point 2 has been completely redrafted.
- Line 106. Please amend “Residua waste”.
The change has been made.
- Figure 2. Please include some sentence such as “a scheme depicting the process of solid waste management is presented in Figure 2”.
The suggested sentence has been entered.
Materials and methods.
- Line 155. The beams should not be introduced as “materials”. It would be better to indicate that, for the tests, concrete beams were casted containing CEM I with a partial substitution of 30% slag. In these lines, the existence of the control (reference) samples without slag should be introduced.
It was introduced.
- Please provide data of “standard sand”.
The sand properties are acc. EN standard.
- Line 163. Please provide another word for “realized”.
Description of method and used materials were generally modified according to the Reviewer’s suggestions.
- Line 164. Up to that moment, slag has never been referred to as “gravel”. Please justify or rewrite.
Thank you. An error has crept in, and word deleted changed.
- Lines 170-171. Cast of the 40x40x160 mm beams.
It was modified in the article according to suggestion.
- Lines 176-177. This step should be rewritten. A possibility is to state it as “Tensile and compressive strength evaluation through destructive testing of the beams. The size of the beams should be introduced when they are cast, not later on in the text. Also, the test methods should be clearly stated regarding compressive and tensile testing. Was compressive and tensile strength estimated from beam tests? If so, please explain.
The compressive strength and tensile strength were evaluated acc. EN standard requirement (mentioned in the article). The unnecessary description of the methodology the test was eliminated.
- Lines 179-190. Is independent that testing from the beams? If so, please introduce it in the materials section as part of the characterization of the slag.
This characterisation was introduced.
- Lines 191-193. When was that testing conducted? Please organize information so it can be easily followed as a standard procedure.
It was introduced.
- Lines 214-220. This is a restatement of what should have been introduced in Lines 170-171. Please check coherence and discourse of all this section. Arguably, some information in lines 170-193 is redundant or should be reorganized.
All Reviewer’s suggestions were introduced to the article.
Results and discussion.
- Line 263. Please check slug/slag. In “so the size recommended …” please rewrite. I suggest it could be “so the slag meets the requirements for GBFS according to the standard…”.
Suggested changes have been made.
- Line 326. Please rewrite. Does crushed concrete refer to portions of the specimens after mechanical characterization? Please, specify the age of maturation in advance, not in line 336.
Suggested changes have been made.
- In order to make the paper is easier to follow, please consider reordering the results so that the fresh properties are discussed before the crushed properties, which were tested after mechanical characterization at 28 days.
Suggested changes have been made.
- Line 390. Please rewrite the sentence for more conventional syntactic order.
It was rewritten.
- Lines 390-424. These statements should be presented in the introduction, to justify the research. If some of this content is included in the discussion section, it must clearly compare the general results with the results from the conducted tests.
This part of the text was moved to the point of introduction separately.
Conclusions.
- Line 427. Please replace “my” by “may”.
As suggested by the Reviewer, it has been changed.
- Line 438. Why previous?.
It was eliminated
Round 2
Reviewer 3 Report
A considerable effort has been made in order to improve the overall quality and presentation of the paper. I believe that your research is worth publishing in its current form.